# Hotspot on 18F-FET PET/CT to Predict Aggressive Tumor Areas for Radiotherapy Dose Escalation Guiding in High-Grade Glioma

**DOI:** 10.3390/cancers15010098

**Published:** 2022-12-23

**Authors:** Bastien Allard, Brieg Dissaux, David Bourhis, Gurvan Dissaux, Ulrike Schick, Pierre-Yves Salaün, Ronan Abgral, Solène Querellou

**Affiliations:** 1Nuclear Medicine Department, University Hospital, 29200 Brest, France; 2UFR Médecine, University of Western Brittany (UBO), 29200 Brest, France; 3GETBO UMR U_1304, Inserm, University of Western Brittany (UBO), 29200 Brest, France; 4Radiology Department, University Hospital, 29200 Brest, France; 5Radiation Oncology Department, University Hospital, 29200 Brest, France; 6LaTIM, INSERM 1101, 29200 Brest, France

**Keywords:** PET/CT, 18F-FET, high grade glioma, hotspot, radiation boost

## Abstract

**Simple Summary:**

For the treatment of high-grade gliomas, radiolabeled amino acid PET/CT could allow for a better tumor delineation for radiotherapy planning and to target aggressive tumor areas for radiotherapy dose escalation guiding. The aim of this ancillary study from the IMAGG prospective trial is to demonstrate a spatial similarity between the areas of high uptake on 18F-FET PET/CT before radio-chemotherapy (MTV), the residual tumor on post-therapy NADIR MRI (GTV 2), and the area of recurrence on MRI (GTV 3). These results on 23 patients showed modest similarity indices between MTV, GTV 2, and GTV 3. Nevertheless, their similarities improved in patients who underwent only biopsy or partial surgery. Delineation methods based on TBR ≥ 1.6 and 80–90% SUVmax showing a good agreement in the hotspot concept for targeting standard dose and radiation boost.

**Abstract:**

The standard therapy strategy for high-grade glioma (HGG) is based on the maximal surgery followed by radio-chemotherapy (RT-CT) with insufficient control of the disease. Recurrences are mainly localized in the radiation field, suggesting an interest in radiotherapy dose escalation to better control the disease locally. We aimed to identify a similarity between the areas of high uptake on O-(2-[18F]-fluoroethyl)-L-tyrosine (FET) positron emission tomography/computed tomography (PET) before RT-CT, the residual tumor on post-therapy NADIR magnetic resonance imaging (MRI) and the area of recurrence on MRI. This is an ancillary study from the IMAGG prospective trial assessing the interest of FET PET imaging in RT target volume definition of HGG. We included patients with diagnoses of HGG obtained by biopsy or tumor resection. These patients underwent FET PET and brain MRIs, both after diagnosis and before RT-CT. The follow-up consisted of sequential brain MRIs performed every 3 months until recurrence. Tumor delineation on the initial MRI 1 (GTV 1), post-RT-CT NADIR MRI 2 (GTV 2), and progression MRI 3 (GTV 3) were performed semi-automatically and manually adjusted by a neuroradiologist specialist in neuro-oncology. GTV 2 and GTV 3 were then co-registered on FET PET data. Tumor volumes on FET PET (MTV) were delineated using a tumor to background ratio (TBR) ≥ 1.6 and different % SUVmax PET thresholds. Spatial similarity between different volumes was performed using the dice (DICE), Jaccard (JSC), and overlap fraction (OV) indices and compared together in the biopsy or partial surgery group (G1) and the total or subtotal surgery group (G2). Another overlap index (OV’) was calculated to determine the threshold with the highest probability of being included in the residual volume after RT-CT on MRI 2 and in MRI 3 (called “hotspot”). A total of 23 patients were included, of whom 22% (*n* = 5) did not have a NADIR MRI 2 due to a disease progression diagnosed on the first post-RT-CT MRI evaluation. Among the 18 patients who underwent a NADIR MRI 2, the average residual tumor was approximately 71.6% of the GTV 1. A total of 22% of patients (5/23) showed an increase in GTV 2 without diagnosis of true progression by the multidisciplinary team (MDT). Spatial similarity between MTV and GTV 2 and between MTV and GTV 3 were higher using a TBR ≥ 1.6 threshold. These indices were significantly better in the G1 group than the G2 group. In the FET hotspot analysis, the best similarity (good agreement) with GTV 2 was found in the G1 group using a 90% SUVmax delineation method and showed a trend of statistical difference with those (poor agreement) in the G2 group (OV’ = 0.67 vs. 0.38, respectively, *p* = 0.068); whereas the best similarity (good agreement) with GTV 3 was found in the G1 group using a 80% SUVmax delineation method and was significantly higher than those (poor agreement) in the G2 group (OV’= 0.72 vs. 0.35, respectively, *p* = 0.014). These results showed modest spatial similarity indices between MTV, GTV 2, and GTV 3 of HGG. Nevertheless, the results were significantly improved in patients who underwent only biopsy or partial surgery. TBR ≥ 1.6 and 80–90% SUVmax FET delineation methods showing a good agreement in the hotspot concept for targeting standard dose and radiation boost. These findings need to be tested in a larger randomized prospective study.

## 1. Introduction

High-grade glioma (HGG) has a poor prognosis, especially glioblastoma with a median survival of less than 10 months [1,2]. The standard first-line therapy is a maximal safe tumor resection followed by radio-chemotherapy (RT-CT) according to the STUPP protocol [3]. Despite recent advances in diagnostic imaging, molecular biology, surgical and radiotherapy techniques, 5-year glioblastoma survival has only increased from 4% to 7% since 1975 [4].

To preserve healthy brain parenchyma and reduce the risk of radionecrosis [5], radiation doses must be limited. These dose constraints could explain why recurrence occurs most often within the radiation field [6]. Recent advances in treatment such as stereotactic RT, intensity modulated radiation therapy (IMRT) or radiosurgery make it possible to deliver a high radiation dose to a localized target. Even the interest of a RT dose escalation beyond 60 Gy has not been proven on large cohorts, studies show interesting results on small series [7,8,9,10,11,12,13]. The use of conventional magnetic resonance imaging (MRI) in radiotherapy planning with gadolinium-enhanced T1-weighted (T1-Gd) and T2-weighted fluid-attenuated inversion recovery (T2-FLAIR) sequences provide only partial information on tumor extension [14]. 

The hotspot concept in positron emission tomography/computed tomography (PET) has already been studied in other solid cancers, showing a significant spatial similarity between high 18-Fluorodesoxyglucose (FDG) uptake area on pre-RT and tumor recurrence scans, mainly in lung, esophagus, and rectal cancers and, less frequently in head and neck squamous cell carcinoma [15]. However, FDG is less used in neuro-oncology because of its physiological uptake by healthy brain parenchyma. In this context, radiolabeled amino acids such as 18F-fluoro-ethyl-L-tyrosine (FET) have been developed, showing better performances in the diagnosis of HGG. The main advantages of FET are a half-life well suited to clinical use, an uptake based predominantly on increased transport via both LAT1 and 2 system without metabolism into the cells neither incorporation into proteins [16], a high in vivo stability, and a high uptake in tumor cells and a low uptake in healthy and inflammatory areas [17,18,19]. Thus, molecular imaging could help define a more specific volume at high risk of recurrence for a modified radiotherapy protocol. The objective would be to guide a radiation boost based on high FET uptake area called a “hotspot”. 

The aim of this study was to assess the spatial similarity between the FET hotspot on pre-RT-CT PET and both residual disease or recurrence tumor volumes. 

## 2. Materials and Methods

### 2.1. Population

This is an ancillary study of the single-center prospective IMAGG trial assessing the interest of FET PET imaging in RT target volume definition of HGG (NCT03370926) [20]. We included patients with diagnosis of HGG obtained by biopsy or tumor resection. These patients underwent both FET PET and brain MRI after diagnosis and before RT-CT, according to the STUPP protocol. Post-treatment follow-up of patients consisted of sequential MRI every 3 months. 

This study was approved by the institutional review board of the University Hospital of Brest (N°2016.CE14). Written informed consent was obtained from all patients.

Patients older than 18 years old with HGG (grade 3 or 4 according to 2016 World Health Organization [21]) and a performance status score ≤ 2 were considered eligible for enrollment. The main exclusion criteria included pregnancy or breastfeeding, contraindications to FET PET and/or MRI, and previous encephalic radiotherapy. Exclusive and/or adjuvant therapy was determined by a multidisciplinary team (MDT).

### 2.2. Imaging Protocol

#### 2.2.1. MRI

MRI scans were performed on a 3T Achieva dStream (Philips healthcare©, Milano, Italia), a 1.5 T Magnetom Avanto Fit (Siemens healthineers©, Erlangen, Germany) or a 1.5 T Optima (General Electric Medical Systems©, Chicago, IL, USA) system. MRI sequences included T1-weighted post-contrast agent T1-Gd (Gd-DTPA; 0.1 mmol/kg body weight) and T2-FLAIR images. 

The first MRI (MRI 1) corresponded to the one performed after biopsy/surgery and before RT-CT. The NADIR MRI (MRI 2), determined by a neuro-oncology radiologist, corresponded to the one with the lowest contrast-enhancement (CE) volume in the post-treatment follow-up. The MRI 3 corresponded to diagnosis of progression validated by a MDT according to the RANO criteria [22]. The design of the study is shown in Figure 1.

#### 2.2.2. FET PET

PET imaging was performed on a Biograph mCT PET system (Siemens Healthineers©, Knoxville, TN, USA). 

For attenuation correction, a low-dose CT scan was performed without iodine contrast. CT acquisition parameters were 16 × 1.2 mm, pitch 0.55 with automatic kVp and mAs modulation. CT reconstruction parameters were slice thickness 3/3 mm, convolution kernel H31s, field of view 500 mm for attenuation correction, and slice thickness 2/1.2 mm, convolution kernel J30s, safire 3, field of view 300 mm for reading. After CT examination, the acquisition was centered on the head and consisted of 40 min dynamic acquisition after the intravenous injection of 3 MBq/kg. PET dynamic reconstructions were performed with 10 × 4 min frames, the reconstruction algorithm was 3DOSEM + TOF + PSF (TrueX) with 200² matrix, zoom2, 2 iterations, 21 subsets, gaussian post filter 2 mm. A single static FET PET frame was obtained by sum 20–40 min. 

All patients fasted for at least 4 h before PET, as per the European Association of Nuclear Medicine (EANM) guidelines for brain tumor imaging using labelled amino acid analogues [23]. 

The delay between neuropathological confirmation obtained by biopsy or tumor resection and RT-CT initiation should not exceed 1 month, and the delay between MRI 1 and FET PET should not exceed 14 days.

### 2.3. Target Volume Delineation on MRI

Tumor volumes delineation was performed using T1-Gd sequences on MRI 1 (GTV 1), on MRI 2 (GTV 2) to avoid pseudoprogression phenomena, and on MRI 3 (GTV 3). 

Residual tumor on MRI 2 represented the percentage of tumor volume persisting after RT-CT and was calculated according to the following formula:(1)$$\frac{\mathrm{GTV 2}}{\mathrm{GTV 1}}\times 100$$

These contours were made semi-automatically using the MIM software (MiM software Inc. Cleveland, OH, USA) in positioning a volume of interest (VOI) upon the tumor to be segmented. A segmentation tool allowing to include only CE volumes and to exclude necrosis areas and resection cavities was then applied for the tumor delineation, visually adjusted by the neuro-oncology radiologist. An explanatory diagram is shown in Figure 2. 

### 2.4. Target Volume Delineation on FET PET

Tumor volumes delineation on FET PET was made using two different methods based on a semi-quantitative index called standard uptake value (SUV):

SUVmax represents the highest radiotracer uptake in one voxel of a volume of interest (VOI).

% SUVmax method = Application of several relative SUVmax thresholds, defined as a three-dimensional contour around voxels equal to or greater than x% (x = 30, 40, 50, 60, 70, 80, and 90) of tumor SUVmax. The 10 and 20% SUVmax thresholds were not included in the data analysis due to outlier results (Figure 3). 

Tumor to Background Ratio (TBR) ≥ 1.6 method = TBR ≥ 1.6 threshold, as already recommended in the literature to define gross tumor volume in FET PET studies [23,24]. The TBR was defined as the tumor SUVmax corrected by the mean background SUV contained in the contralateral normal brain tissue, including white and grey matter, in a crescent-shaped VOI (called “banana”) resulting from the summation of 6 consecutive ROIs of 20 mm diameter [25].

### 2.5. Co-Registration

T1-Gd MRI 1, T1-Gd MRI 3, and FET PET images were elastically co-registered to the T1-Gd MRI 2 and considered as the reference. In absence of MRI 2, T1-Gd MRI 3 was elastically co-registered to T1-Gd MRI 1. 

The MRI–MRI co-registrations were performed in two steps: the first step consisted of a rigid co-registration using an algorithm constrained in a VOI that included the tumor and where the similarity measure was performed. The second step consisted of a deformable co-registration, performed using the VoxAlign Deformation Engine, a constrained intensity based free-form deformable registration algorithm [26]. 

Because the MRI-PET deformable registration was not allowed, PET was first co-registered to MRI 1 (rigid registration), assuming that the delay between MRI 1 and PET was short enough to avoid a spatial decorrelation. Then, the MRI 1 to MRI 2 deformable registration matrix was applied on PET images.

### 2.6. Spatial Similarity Coefficients

The different SUVmax thresholds and the TBR ≥ 1.6 of FET PET were compared with the GTV 2 (Figure 4) and GTV 3 (Figure 5). 

To assess spatial similarities of the different volumes, the dice similarity coefficient (DICE), the Jaccard similarity coefficient (JSC), and the overlap fraction (OV) were calculated [27]. 

Their indices are widely used to compare delineated volumes obtained with different methods or by multiple investigators. Their values vary between 0, if the volumes are completely disjointed, and 1, if the volumes match perfectly in size, shape, and location (Figure 6). 

DICE was defined as followed: $$\mathrm{DICE}=\frac{2(\mathrm{MTV}\cap \mathrm{GTV})}{\mathrm{MTV}+\mathrm{GTV}}$$

JSC was calculated as followed: $$\mathrm{JSC}=\frac{\mathrm{MTV}\cap \mathrm{GTV}}{\mathrm{MTV}\cup \mathrm{GTV}}$$

OV was defined as followed: $$\mathrm{OV}=\frac{\mathrm{MTV}\cap \mathrm{GTV}}{\min(\mathrm{MTV}\times \mathrm{GTV})}$$

### 2.7. Hotspot Concept

To assess the best spatial similarity rate of the MTV before RT-CT with GTV 2 and GTV 3, the index (OV’) defined as follow was used: $${\mathrm{OV}}^{\prime }=\frac{\mathrm{MTV}\cap \mathrm{GTV}}{\mathrm{MTV}}$$

The goal was to have the “hotspot” fully included in the persistent volume after RT-CT (GTV 2) and the progression volume (GTV 3) to guide a radiation boost (Figure 7).

### 2.8. Statistical Test

The analysis of spatial similarity indices obtained between PET and MRI volumes was performed according to two groups. The first group (G1) included patients who underwent biopsy alone or partial surgery. The second group (G2) included patients who had undergone total or subtotal surgery. The resection was termed partial, subtotal, and total if <90%, ≥90%, and 100% of CE, respectively, was resected.

The differences between the respective DICE, JSC, OV, and OV’ indices for these two groups were tested using a Student t-test with the hypothesis that the spatial similarity was better in G1. All statistical analyses were performed with XLStat 2019 software (Addinsoft©, Paris, France). A *p* value ≤ 0.05 was considered statistically significant.

The Landis and Koch scale was used to classify the quality of overlap: 0–0.2, poor agreement; 0.21–0.40, fair agreement; 0.41–0.60, moderate agreement; 0.61–0.80, good agreement; and 0.81–1.00, very good agreement [29]. 

## 3. Results

### 3.1. Patient Characteristics

Among the 30 patients with newly diagnosed HGG prospectively included between November 2016 and December 2018 in the IMAGG study [20], 7 were excluded from our analysis for different reasons listed in the flowchart (Figure 8).

Finally, 23 patients (sex ratio 14M/9F) with mean age at diagnosis of 59y (range 38 to 74) were analyzed. All HGGs were astrocytomas with a majority of grade IV (87%). The distribution in the G1 and G2 groups was globally equivalent (52% versus 48%, respectively). Molecular biology revealed MGMT methylation in 13 patients. All patient characteristics are summarized in Table 1.

### 3.2. Impact of Radio-Chemotherapy

In our population, five patients did not have MRI 2 due to a disease progression on MRI performed after RT-CT. Among the remaining 18 patients, 13 reached a reduction in CE between MRI 1 and MRI 2. A mean decrease of −28.4% (range −98.4% to +100%) was found. 

The variations between GTV 1 and GTV 2 are presented in Table 2 for each patient according to their characteristics. 

### 3.3. Spatial Similarity between PET (MTV) and MRI Tumor Volumes (GTV)

All results of the mean spatial similarity indices (DICE, JSC, and OV) between MTV according to different delineation methods and respective GTV 2 and GTV 3 are summarized in Table 3 and Table 4. Six patients had distant recurrences (out of field) without local recurrence and these patients were excluded from the analysis of spatial similarity between MTV and GTV 3. 

The best similarity between MTV and GTV 2 (good agreement) was found in group G1 using a 30% SUVmax delineation method (OV = 0.781). The best compromise in indices comparison between the G1 and G2 groups was found with the TBR ≥ 1.6 method (DICE 0.418 vs. 0.207, JSC 0.287 vs. 0.127, and OV 0.735 vs. 0.477, *p* < 0.05, respectively).

The best similarity between MTV and GTV 3 (good agreement) was found in group G1 using a TBR ≥ 1.6 threshold delineation method (OV = 0.757). The best compromise in indices comparison between the G1 and G2 groups was also found with the TBR ≥ 1.6 method (DICE 0.488 vs. 0.233, JSC 0.339 vs. 0.144, and OV 0.757 vs. 0.434, *p*< 0.05, respectively). 

### 3.4. Hotspot Concept

All results of the mean similarity indices (OV’) for FET hotspot analysis according to different delineation methods are summarized in Table 5.

The maximal similarity (good agreement) between the FET hotspot and the GTV 2 was found in the G1 group using a 90% SUVmax delineation method and higher than those of the G2 group with a trend of statistical significance (OV’ = 0.67 vs. 0.38, respectively, *p* = 0.068). The maximal similarity (good agreement) between the FET hotspot and the GTV 3 was found in the G1 group using a 80% SUVmax delineation method and was significantly higher than those in the G2 group (OV’= 0.72 vs. 0.35, respectively, *p*= 0.014).

## 4. Discussion

Our study aimed to prove that FET hotspot on initial PET could identify tumor areas at high risk for relapse in HGG in comparing with tumor volumes on both MRI NADIR (MRI 2) and MRI at progression (MRI 3). 

First-line treatment currently based on maximal surgery followed by concomitant RT-CT and adjuvant CT is always insufficient with an incomplete control of disease and majority recurrences occurring within the treated high-dose volume [6]. With such a background, the key issue is improving patient management and one of the challenges relies on RT dose escalation. Beyond MRI-based gross tumor volume (GTV) delineation, a metabolic tumor volume (MTV) may be also defined by functional PET imaging. Histopathological and postmortem series demonstrated the limitations of conventional MRI in defining the extent of glioma [30,31]. Thus, guidelines published by RANO/EANO group confirmed the interest of molecular imaging [23]. To the best of our knowledge, our study is one of the first using this PET tracer to demonstrate such objective.

This is an ancillary study from the IMAGG prospective trial assessing the interest of FET PET imaging in RT target volume definition of HGG. Two different classes of PET radiotracers have been used in neuro-oncology, historically the FDG to explore glucose metabolism and recently amino-acid tracers; however, due to a high physiological uptake in normal brain, a lower signal-to-noise ratio of brain tumors and a risk of false positive with post-therapeutic inflammatory sequelae, the use of FDG has decreased. The use of radiolabeled amino acids, especially FET or 18F-DOPA (DOPA), has grown in recent years. Thereby, FET provides metabolic data for brain tumor management [32] with a higher specificity than FDG [33]. The main advantages of FET over DOPA are its high in vivo stability and its uptake based mainly on increased transport through the L-amino acid transport system without metabolism into the cells or incorporation into the proteins [34]. Moreover, by using dynamic data, the diagnostic performances for FET can be improved [35,36,37] and are currently studied for DOPA [23] whereas kinetic analyses are difficult to achieve for 11C-MET due to its short radioactive half-life [17]. 

As suggested by previous articles, MTV delineated on amino-acid radiolabeled PET imaging are different, mostly larger and more pertinent regarding neuropathological findings than GTV realized on MRIs, especially in newly diagnosed glioblastoma [38,39]. Girard et al. confirmed that the combination of DOPA PET imaging with multimodal MRI enlarged the delineation volumes and enhanced overall accuracy to detect high-grade areas in a series of 16 patients (4 grade II, 6 grade III, 6 grade IV) with 38 biopsy samples. They underlined for three patients an intra-tumoral heterogeneity with coexisting low and high grade tumor subregions [40]. Song et al. demonstrated that MTV FET PETs were significantly larger than GTV MRI (77.84 ± 51.74 mL vs. 34.59 ± 27.07 mL, *p* < 0.05) with a higher similarity in histopathology in 31 gliomas. Thus, of 21 biopsy samples targeted on areas with increased FET uptake, all were neuropathologically confirmed as tumor tissue, only 13 revealed a contrast enhancement on MRI [41]. In the same way, Filss et al. revealed that in 56 WHO grades II/IV gliomas that metabolically active tumor volume delineated on FET PET was significantly larger than cerebral blood tumor volume calculated on perfusion-weighted MRI (24.3 ± 26.5 mL vs. 8.9 ± 13.9 mL, *p* < 0.001) [42]. So, GTV being underestimated by MRI, the capability to better define tumor extent with functional PET imaging may be useful to modify the radiation therapy. Actually, studies focused on radiolabeled amino acids PET-based radiotherapy planning with a standard dose demonstrated a predominance of local relapse within the high-dose-treated volume [11,12]. This suggests that the radiation dose delivered is insufficient for local tumor control and justifies moving towards a dose-escalation approach; however, dose escalation is limited by the tolerance of surrounding tissues and the radiation-induced toxicity as necrosis. The possibility to define a more specific volume with a high risk of residual or recurrent disease may be useful to guide radiotherapy with the rising of innovative techniques such as stereotactic radiotherapy or IMRT.

Thus, some authors also previously suggested that pre-treatment FET PET may be useful to predict tumor recurrence in HGG. Indeed, in a series of 44 glioblastomas, Piroth et al. showed that postoperative pre-RT tumor volume in FET PET was an independent prognostic factor of overall survival (OS) and disease free survival (DFS) (OS 20.0 vs. 6.9 months; DFS 9.6 vs. 5.1 months, *p* < 0.001) using a cut-off of 25 mL [43]. Lundemann et al. developed from an analysis of 16 glioblastomas a predictive model of recurrence including both FDG and FET tumor volumes and radiomic features with an AUC of 0.77 [44]. These emphasized again that FET PET may be appropriate to guide dose escalation.

Based on the previous data, many studies aimed to predict the location of residual or relapsed tumors using FDG PET functional imaging. In a systematic review including nine studies, Abgral et al. identified all articles reporting on a similarity between high FDG uptake called “hotspot” on pretreatment PETs (PET_A_) and sites of local recurrence of several solid tumors on PETs after radiotherapy (PET_R_) [15]. The authors concluded that similarity between FDG hotspot on PET_A_ and areas of local recurrence on PET_R_ showed good to excellent agreement (ranging from 0.60 to 0.93) for lung, esophageal, and rectal carcinomas. They highlighted lower agreement for head and neck cancer, weight loss and tissue distortion after RT-CT affecting the anatomical landmarks and increasing the risk of mis-registrations [45], even if PET_A_ and PET_B_ were acquired in the same position and registered with an elastic method [46]. Nevertheless, no studies were reported on HGG.

As widely used in such FDG hotspot pre-radiotherapy PET studies, we have chosen dice, Jaccard, and overlap fraction indices to compare different volumes extracted from the PET and MRI data. We also arbitrary tested another index that we called overlap fraction’ (OV’) defined as the intersection between GTV and MTV divided by the MTV. In our opinion, it makes more sense to report MRI/PET intersection systematically to the PET volume, in this perspective of PET-based RT dose escalation. We found modest similarity indices between pre-treatment MTV on FET PET and both residual/recurrent disease GTV on MRI with dice and Jaccard coefficients, but their results are good for the overlap fraction indices. There was a significant improvement in our G1 group of patients who underwent only biopsy or partial surgery. This underlines that there are postoperative tissue distortions in the braincase within the porencephalic cavity, altering the anatomical landmarks.

We applied a SUV-based method by testing different SUVmax thresholds to delineate MTV. This is a simple, semi-automated, and routinely used measurement with a high reproducibility [47]. In addition, we chose to test a TBR threshold ≥ 1.6 as it was also validated by a previous HGG biopsy-controlled study in FET PET. Similarly, one of its advantage is the high inter-observer agreement for tumor volume delineation [48]. In our series, these two approaches finally proved to be complementary in that the highest spatial similarity was achieved with a TBR threshold ≥ 1.6 (OV = 0.735) and the best intersection with the FET hotspot was achieved with 90% SUVmax and 80% SUVmax thresholds on MRI 2 (OV’ = 0.67) and MRI 3 (OV’ = 0.72), respectively, in the biopsy or partial surgery group.

Our study has some limitations. Firstly, our population was relatively low (*n* = 23) but all demographic characteristics were quite similar to the literature excepted for a lower percentage of maximal surgery [49]. Secondly, for this study, the WHO 2016 classification was used instead of the updated WHO 2021 classification, as it is an ancillary study of the prospective IMAGG study conducted in 2016 and all the elements to update the tumor grades weren’t available. Then, we used the RANO criteria to make the diagnosis of progression disease (PD). Some patients could have been declared in PD due to a deterioration in performance status, a decision to introduce corticosteroid therapy, or a change in the T2 FLAIR hypersignal on MRI. However, of the 23 patients, only 1 achieved a PD without a CE increase on MRI, which probably has a limited impact on our tumor volume similarity analysis based on T1-weighted sequences. Thirdly, we chose to delineate tumors using a high reproducible semi-automated method on a single MRI sequence [48]. We decided to exclude T2-FLAIR-weighted tumor volumes in our analysis because this sequence does not allow differentiation between tumor infiltration and perilesional edema and therefore lacks specificity. For this reason, some authors of the RANO criteria also suggest to abandon this sequence for HGG treatment response assessment [50,51]. However, other MRI sequences such as T2 FLAIR may be evaluated, especially for non-enhancing tumors. Finally, our method of MRI-PET co-registration may be criticized but it underlines the potential difficulty of such process even for brain imaging despite appropriate restraint system during both imaging. We proved in a previous hotspot study of head and neck cancers in FDG PET that the use of this step by step deformable registration method limited the impact of post-RT-CT remodeling and improved the similarity indices of studied volumes [46]. In a future perspective, combined PET/MRI systems, which provide an interesting way in dedicated brain imaging, may optimize this co-registration and thus improve spatial similarity indices [41,42,52,53]. Finally, FET may be expensive and is not available in all countries. The radio-labelled amino acids (FET, DOPA, and MET) seem to have equivalent performances in neuro-oncology and the therapeutic strategy proposed in this study could be applied to DOPA and MET depending on the availability and use of the centers [17]. 

Thus, radio-labeled amino acid PET could be used in the future in radiotherapy planning to guide dose escalation on at-risk subpopulations. The interest of dose escalation in HGG has not yet been clearly demonstrated. Dose-escalation studies with high doses to regions of MRI contrast enhancement were unsuccessful in improving survival in glioblastoma [54,55]. Dose-escalation studies in HGG used conventional MRI sequences (T1-GD and T2-FLAIR) for radiotherapy planning, which could explain in part the disappointing results. Amino acid PET/CT could allow better targeting of tumor limits as the tracer transport is independent from the blood-brain barrier breakdown especially for non-enhancing tumor and it’s correlated with grade and cellularity [56]. Recently, Laack et al. in a phase-2 trial studied the safety and efficacy of biologically guided dose-escalated radiation therapy using DOPA in patients with glioblastoma. In this study, a boost radiation of 76 Gy was performed in areas with a TBR ≥ 2. A TBR of 1.2 to 2.0 was used to guide the delineation of 51 Gy metabolic target volume. This study showed that dose escalation guided by DOPA PET appeared to be safe, and it significantly improved progression free survival in MGMT un-methylated glioblastoma and overall survival in MGMT methylated patients [11].

Our study reveals that FET PET before RT-CT is better at predicting areas at risk in patients who have not undergone maximalist surgery (biopsy alone or partial surgery). This may be explained by a non-specific uptake of FET on areas of inflammation and parenchymal remodeling after neurosurgical intervention. The hotspot could then be located both on the inflammation zone and on the residual tumor. Therefore, in our opinion, this therapy approach of dose escalation guided by FET PET in HGG seems much more appropriate in patients whose total or subtotal surgery is not possible due to an altered general condition or a difficult-to-access tumor location. These patients usually have a severe prognosis with limited therapeutic proposals. This study aims to introduce a novel therapeutic perspective for these patients.

## 5. Conclusions

In conclusion, FET PET-guided RT dose-escalation approach in high-grade gliomas may be more appropriate for patients for whom total or subtotal surgery is not possible. In this case, our results suggested using a TBR threshold ≥ 1.6 to determine the MTV in which the standard RT dose will be delivered before applying a 80% or 90% SUVmax threshold to the defined hotspot at risk of residual/recurrence disease in which the irradiation boost will be delivered.

## Figures and Tables

**Figure 1 cancers-15-00098-f001:**
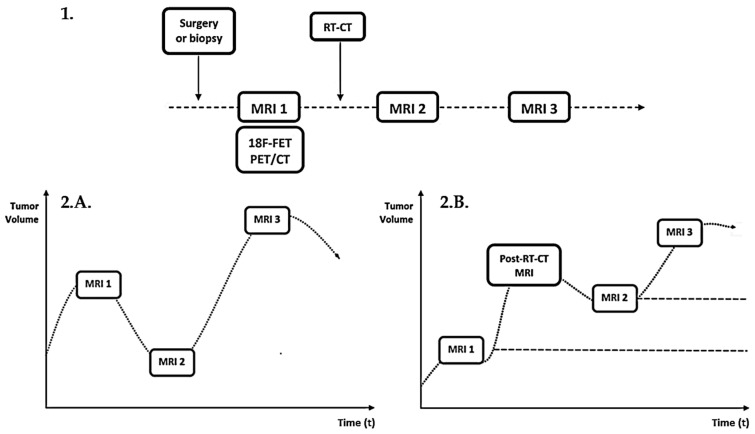
**1**: Study design. **2**: Diagram of the MRI timeline. **A**. An example with a smaller tumor volume on MRI 2 than on MRI 1. **B**: An example with a higher tumor volume on MRI 2 than on MRI 1 (MRI 2 represent the smallest tumor volume in the post-RT-CT follow-up).

**Figure 2 cancers-15-00098-f002:**
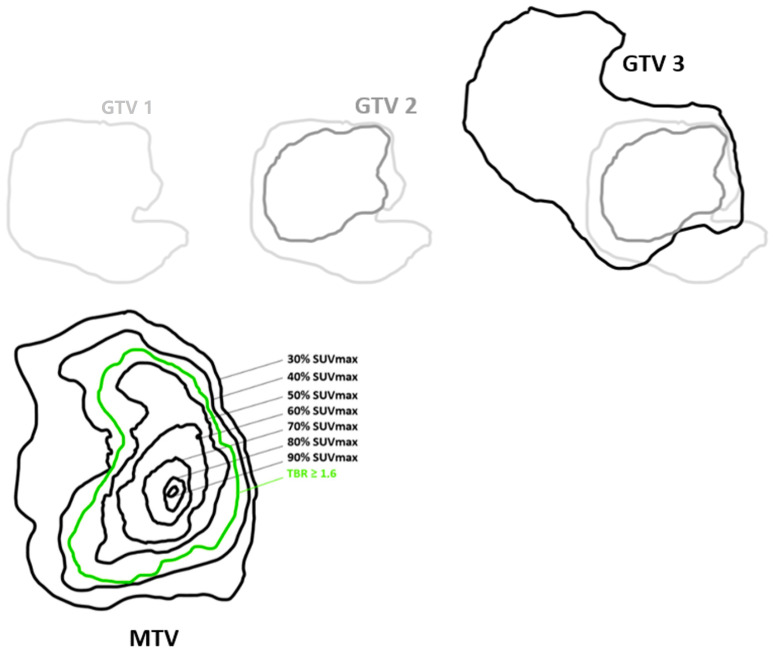
Simplified schematic representation of tumor delineation on pre-RT-CT MRI (GTV 1), post-therapy NADIR MRI (GTV 2) and progression MRI (GTV 3) and the different thresholds of the tumor lesion on FET PET before RT-CT (MTV).

**Figure 3 cancers-15-00098-f003:**
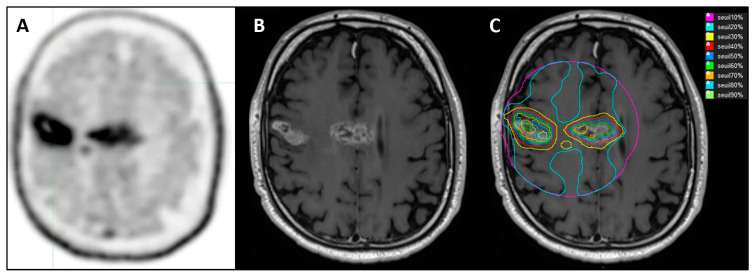
A 68-year-old patient with multifocal glioblastoma in the right frontal area and corpus callosum. (**A**). FET PET after surgical biopsy and before RT-CT. (**B**): NADIR MRI (MRI 2) after RT-CT in T1-GD sequence 3 months after FET PET. (**C**): Different SUVmax thresholds on MRI 2 after co-registration.

**Figure 4 cancers-15-00098-f004:**
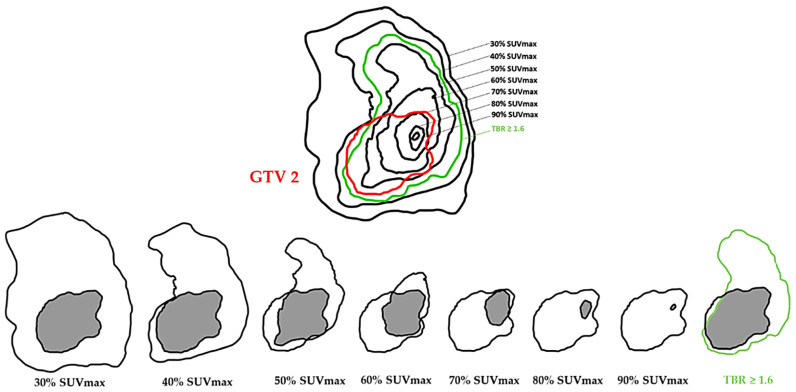
Simplified schematic representation of comparison of tumor volumes on FET PET before RT-CT with tumor volume on post-therapy NADIR MRI 2 (GTV 2).

**Figure 5 cancers-15-00098-f005:**
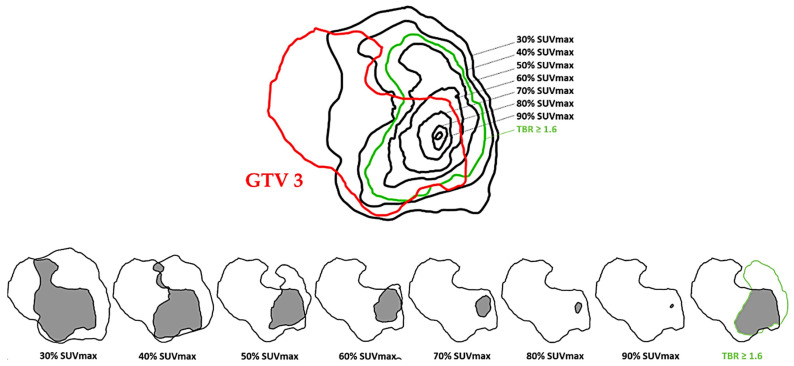
Simplified schematic representation of comparison of tumor volumes on FET PET before RT-CT with GTV 3.

**Figure 6 cancers-15-00098-f006:**
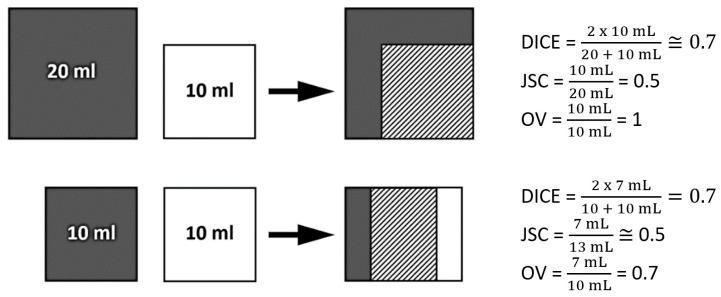
Simplified schematic representation of the different spatial similarity indices (DICE, JSC and OV) modified from Lohmann et al. [28].

**Figure 7 cancers-15-00098-f007:**
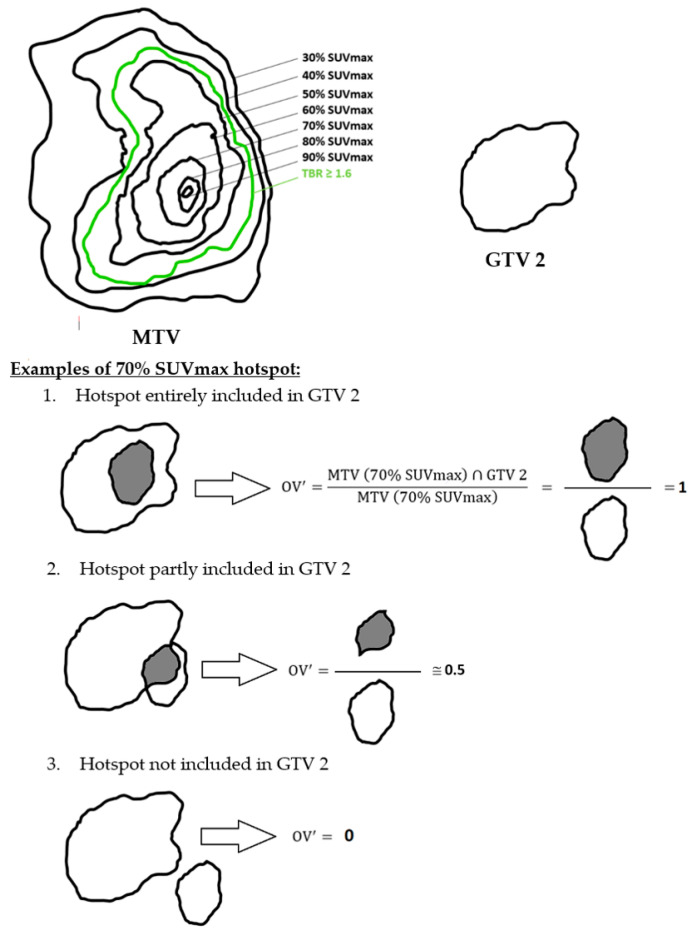
Simplified schematic representation of the hotspot concept using the 70% SUVmax threshold with GTV 2.

**Figure 8 cancers-15-00098-f008:**
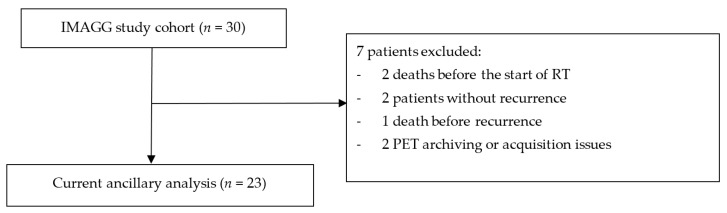
Flowchart.

**Table 1 cancers-15-00098-t001:** Patient characteristics.

	Mean/Number	Percentage/ Range
Age at diagnosis	59	38–74
Sex		
Male Female	14 9	60.9% 39.1%
Cell origin		
Astrocytoma Oligodendroglioma	23 0	100% 0%
Histology		
Grade III Grade IV	3 20	13% 87%
Neurosurgical intervention before inclusion:		
Total surgery (G2) Subtotal surgery (G2) Partial surgery (G1) Biopsy alone (G1)	8 3 3 9	34.8% 13% 13% 39.1%
Time between MRI 1 and MRI 2	255	71–938
Time between MRI 1 and MRI 3	330	48–1033
Molecular biology:		
IDH mutation Co-deletion 1p/19q MGMT methylation	0 0 13	0% 0% 57%

**Table 2 cancers-15-00098-t002:** Population characteristics and variation in tumor volumes between MRI 1 (GTV 1) and MRI 2 (GTV 2).

N°	Age	Gender	Surgery * (Group)	Histology	Ki67	GTV 1 mL	GTV 2 Ml	Residual tumor (%)	Variation (%)
1	45	M	3 (G2)	IV	20%	6.6	3.8	57.6	−42.4
2	65	M	3 (G2)	IV	0.1%	21.7	NA	NA	NA
3	56	M	1 (G1)	IV	30%	23.6	2.2	9.3	−90.7
4	70	M	0 (G1)	IV	30%	53.0	51.7	97.5	−2.5
5	49	M	0 (G1)	IV	20%	62.5	NA	NA	NA
6	70	F	3 (G2)	IV	15%	11.4	5.4	47,4	−52.6
7	57	F	2 (G2)	IV	30%	3.0	0.1	3.3	−96.7
8	63	M	3 (G2)	IV	0.7%	13.7	12.1	88.3	−11.7
9	70	M	0 (G1)	IV	20%	38.8	NA	NA	NA
10	64	M	1 (G1)	III	10%	0.9	0.95	105.6	+5.6
11	61	F	3 (G2)	IV	0.3%	6.4	0.1	1.6	−98.4
12	74	M	0 (G1)	IV	20%	18.6	5.4	29.0	−71.0
13	53	F	3 (G2)	IV	5%	5.9	3.1	52.5	−47.5
14	61	F	3 (G2)	IV	20%	10.5	21.0	200.0	+100.0
15	64	M	0 (G1)	IV	10%	31.6	32.3	102.2	+2.2
16	47	F	3 (G2)	IV	15%	0.5	0.3	60.0	−40.0
17	44	M	1 (G1)	IV	0.2%	2.0	NA	NA	NA
18	49	M	0 (G1)	III	20%	37.9	26.0	68.6	−31.4
19	68	M	0 (G1)	IV	10%	7.5	8.5	113.3	+13.3
20	38	M	2 (G2)	IV	30%	6.5	10.1	155.4	+55.4
21	63	F	2 (G2)	IV	60%	16.5	1.1	6.7	−93.3
22	58	F	0 (G1)	III	35%	1.4	NA	NA	NA
23	69	F	0 (G1)	IV	40%	38.7	37.3	96.4	−3.6

* 0: Biopsy only/1: Partial surgery/2: Subtotal surgery/3 Total surgery. NA: Not Applicable.

**Table 3 cancers-15-00098-t003:** Mean spatial similarity indices between MRI 2 according to different pre-therapy PET thresholds (5 patients excluded due to none MRI 2).

	G1 = Biopsy or Partial Surgery (*n* = 8)			G2 = Total or Subtotal Surgery (*n* = 10)			Difference between G1 and G2 (*p*=)		
	DICE	JSC	OV	DICE	JSC	OV	DICE	JSC	OV
30% SUVmax	0.362	0.243	**0.781**	0.138	0.079	**0.684**	0.01 *	0.008 *	0.223
40% SUVmax	0.397	0.270	0.722	0.173	0.103	0.570	0.015 *	0.013 *	0.120
50% SUVmax	0.368	0.246	0.676	0.193	0.125	0.462	0.057	0.076	0.047 *
60% SUVmax	0.288	0.181	0.632	0.176	0.114	0.358	0.131	0.188	0.028 *
70% SUVmax	0.196	0.114	0.596	0.136	0.083	0.344	0.225	0.270	0.051
80% SUVmax	0.114	0.068	0.592	0.100	0.060	0.325	0.428	0.432	0.059
90% SUVmax	0.032	0.026	0.670	0.019	0.017	0.379	0.644	0.658	0.068
TBR ≥ 1.6	**0.418**	**0.287**	0.735	**0.207**	**0.127**	0.477	0.024 *	0.019 *	0.012 *

* Statistically significant (Student T test). Bold: Best spatial similarity indices.

**Table 4 cancers-15-00098-t004:** Mean spatial similarity indices between MRI 3 according to different thresholds on pre-therapy PET (6 patients excluded for distant recurrence (out of field) without local recurrence).

	G1 = Biopsy or Partial Surgery (*n* = 10)			G2 = Total or Subtotal Surgery (*n* = 7)			Difference between G1 and G2 (*p*=)		
	DICE	JSC	OV	DICE	JSC	OV	DICE	JSC	OV
30% SUVmax	0.465	0.321	0.755	0.215	0.132	**0.551**	0.009 *	0.010 *	0.046 *
40% SUVmax	0.483	0.331	0.707	0.199	0.126	0.451	0.004 *	0.006 *	0.019 *
50% SUVmax	0.445	0.300	0.670	0.173	0.112	0.400	0.006 *	0.012 *	0.022 *
60% SUVmax	0.338	0.213	0.670	0.141	0.087	0.370	0.018 *	0.028 *	0.021 *
70% SUVmax	0.196	0.113	0.701	0.105	0.061	0.353	0.094	0.113	0.012 *
80% SUVmax	0.078	0.045	0.717	0.078	0.042	0.362	0.503	0.545	0.016 *
90% SUVmax	0.019	0.010	0.681	0.018	0.009	0.367	0.470	0.473	0.061
TBR ≥ 1.6	**0.488**	**0.339**	**0.757**	**0.233**	**0.144**	0.434	0.008 *	0.007 *	0.003 *

* Statistically significant (Student T test). Bold: Best spatial similarity indices.

**Table 5 cancers-15-00098-t005:** Mean ratios of the intersection of MRIs and FET volumes to the FET volume (OV’).

	OV’ (MTV; GTV 2)			OV’ (MTV; GTV 3)		
	G1 (*n* = 8)	G2 (*n* = 10)	Difference (*p*=)	G1 (*n* = 10)	G2 (*n* = 7)	Difference (*p*=)
30% SUVmax	0.31	0.09	0.019 *	0.41	0.17	0.017 *
40% SUVmax	0.4	0.14	0.014 *	0.51	0.21	0.007 *
50% SUVmax	0.46	0.19	0.021 *	0.60	0.24	0.004 *
60% SUVmax	0.5	0.23	0.034 *	0.66	0.25	0.003 *
70% SUVmax	0.54	0.26	0.043 *	0.70	0.28	0.003 *
80% SUVmax	0.59	0.31	0.053	**0.72**	0.35	0.014 *
90% SUVmax	**0.67**	**0.38**	0.068	0.68	**0.37**	0.061
TBR ≥ 1.6	0.33	0.153	0.027 *	0.41	0.21	0.022 *

* Statistically significant (Student T test). Bold: Best spatial similarity indices.

## Data Availability

The data are not publicly available due to the facility’s privacy policy.

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
