# Peer review of "Hotspot on 18F-FET PET/CT to Predict Aggressive Tumor Areas for Radiotherapy Dose Escalation Guiding in High-Grade Glioma"

_cancers, 2022, doi:10.3390/cancers15010098_

Round 1

Reviewer 1 Report

This is an interesting study utilizing a posthoc analysis from an ongoing research protocol. The challenge of defining a biological target for radiation dose escalation in glioblastoma is of significant interest to neuro-oncology community especially with the minimal impact on outcome of systemic therapies in recent RCTs.

I believe the manuscript is of quality for publication without any significant changes.

Some minor comments that could be expanded by the authors:

  1. This paper provides a novel model for target volume design and quantitative use of FET PET imaging data. The major limitation is that the postoperative inflammatory changes recognised on FET in GTV1 may not parallel the conventions of SUVmax and TBR for tumour. Thus there may be an error in determination of metabolic hotspot in GTV1. A sentence explaining this uncertainty could be added.
  2. A major limitation related to the lack of inclusion of non-enhancing tumour and T2 Flair in the model. The authors recognise this limitation in the discussion, and an additional comment may be made to detail the uncertainty in the model created by the increase in CE and FET at post RT assessment.

Author Response

best regards

Reviewer 2 Report

I read with particular interest this article by Allard et al.

The topic is very interesting.

It’s very difficult to understand study methodology. I’ll try to rephrase sentences in M&M section.

I also suggest you verify spelling and English grammar because of many French sounds (ex: spatial corrélation)

I know that authors previous published this article “Radiotherapy target volume definition in newly diagnosed high grade glioma using 18F-FET PET imaging and multiparametric perfusion MRI: A prospective study (IMAGG)”. I wouldn't assume that everyone is familiar with that article.

Some specific comments:

Abstract:

-        I use standard and not reference, first line

-        Line 30: histological is surgical, bioptic, imaging derived…

Introduction:

-        Lines 67-69: I suggest you rephrase this sentence

-        What is the peculiarity in FET PET use?

M&M:

-        Line 92: biopsy or neurosurgical, better define

-        Grade 3 and 4 according to 2016 WHO. Is it possible to define the grade according to new high-grade glioma classifications?

-        What is the base of the math formula in paragraph 2.3?

Results:

-        Where is the recurrence in PET-MR3? In field? Out of field?

-        In abstract the authors allude to necrosis. In results no mention of it?

-        Is there a correlation between necrosis and MRI or PET findinigs?

Discussion:

-        As previous said, the methodology of the study is very hard to understand (also for a radiation oncologist as me). I’ll try to better define the clinical benefit, the clinical new aspects, the advantages of FET PET using. I also will point out the difficult in use this type of imaging.

-        What is the cost?

Author Response

best regards.

Round 2

Reviewer 2 Report

I have no more comments.